# RAVSS: Robust Audio-Visual Speech Separation in Multi-Speaker Scenarios with Missing Visual Cues

## ABSTRACT

While existing Audio-Visual Speech Separation (AVSS) methods primarily concentrate on the audio-visual fusion strategy for two-speaker separation, they demonstrate a severe performance drop in the multi-speaker separation scenarios. Typically, AVSS methods employ guiding videos to sequentially isolate individual speakers from the given audio mixture, resulting in notable missing and noisy parts across various segments of the separated speech. In this study, we propose a simultaneous multi-speaker separation framework that can facilitate the concurrent separation of multiple speakers within a singular process. We introduce speaker-wise interactions to establish distinctions and correlations among speakers. Experimental results on the VoxCeleb2 and LRS3 datasets demonstrate that our method achieves state-of-the-art performance in separating mixtures with 2, 3, 4, and 5 speakers, respectively. Additionally, our model can utilize speakers with complete audio-visual information to mitigate other visual-deficient speakers, thereby enhancing its resilience to missing visual cues. We also conduct experiments where visual information for specific speakers is entirely absent or visual frames are partially missing. The results demonstrate that our model consistently outperforms others, exhibiting the smallest performance drop across all settings involving 2, 3, 4, and 5 speakers.

## CCS CONCEPTS

• **Computing methodologies** → **Artificial intelligence**; • **Information systems** → *Multimedia information systems*.

## KEYWORDS

Audio-visual speech separation, multi-speaker scenarios, missing visual cues

## 1 INTRODUCTION

Humans can selectively concentrate on desired sounds in complex acoustic environments with mixed speech signals from multiple speakers and diverse background noises, which is termed the "Cocktail Party Problem" [4, 8, 15]. Audio-Visual Speech Separation (AVSS) is a task that aims to separate and isolate individual speeches from the overlapped speech mixtures along with corresponding visual cues like speaker's face [14] or lip-movements [31, 32]. AVSS serves as a front-end process for tasks like speech recognition and

*ACM MM, 2024, Melbourne, Australia*
© 2024 Copyright held by the owner/author(s). Publication rights licensed to ACM.
ACM ISBN 978-x-xxxx-xxxx-x/YY/MM
https://doi.org/10.1145/nnnnnnn.nnnnnnn

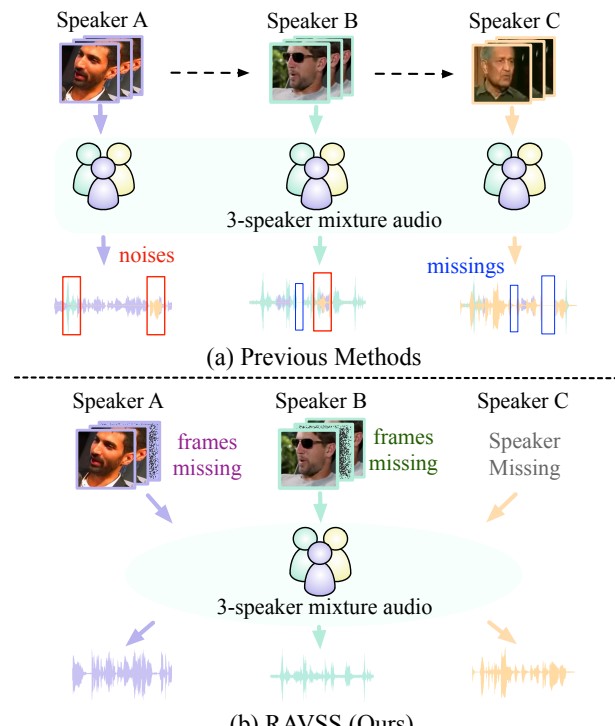

**Figure 1: AVSS task description and the contribution of our work. (a) shows the basic audio-visual speech separation process. It uses the visual cue to extract the corresponding speech from the mixture. The separation process is repeated to separate more speakers. (b) demonstrates our proposed separation process, which addresses the task of separating multiple speakers jointly. It can simultaneously separate multiple speech sources using multiple visual cues, maintaining robustness to missing visual cues.**

speaker diarization, finding applications in hearing aids devices [41] and teleconferencing systems [6, 35].

Recent advancements in AVSS methods [9, 14, 21, 22, 33] have primarily focused on enhancing audio-visual fusion strategies for the two-speaker separation scenario. They typically utilize visual cues as guidance to sequentially separate each corresponding speaker's speech from the same mixture. While this approach yields satisfactory results in separating two speakers without significant performance degradation, it faces challenges when dealing with three or more speakers. In such scenarios, the conventional one-by-one speaker extraction method often leads to numerous instances of missing and noisy segments [7] within each separated speech stream, as depicted in Figure 1(a). As a result, this separation approach exhibits a significant performance drop, as illustrated in

Table 1, indicating its limited capability to differentiate and capture interactions among multiple speakers comprehensively. Several methods [1, 3, 13] have also been developed to address multi-speaker scenarios. L2L [13] employs customized models for each mixture type in each one-by-one speaker extraction process. And Afouras et al. [1, 3] enhance the one-by-one speaker extraction process with specifically designed modules or pre-enrolled speaker embeddings. Furthermore, BFRNet [7] proposes a filter-recovery network as a further post-processing refinement step. However, these methods also fail to address the separation of multiple speakers in a unified way. As illustrated in Figure 1(b), we propose a robust multi-speaker audio-visual separation framework that can separate multi-speakers collaboratively in a single separation process. To achieve this, we suggest two speaker-specific interaction forms: speaker-wise audio-only and speaker-wise audio-visual interaction. The speaker-wise audio-only interaction assesses the relationships of distinctions and connections within each speaker. It effectively mitigates the relatively notable missing segments and eliminates irrelevant parts for each speaker, thereby achieving a more accurate separation of multiple speakers. The speaker-wise audio-visual interaction can further enhance the inter-speaker disparity by providing more evident visual distinctions between speakers. It offers a more detailed refinement to rectify or remove segments that are initially similar between speakers. Incorporating these two speaker-specific interactions enhances the model's robustness, particularly when the number of speakers in the audio mixture increases.

In real-world scenarios, various factors such as a speaker moving off-screen or the camera being turned off can result in missing video information [3, 5, 11, 16, 20, 42]. We additionally evaluate the robustness of our model in handling missing video data, where visual information for specific speakers is absent or where visual frames are partially missing. Firstly, we expand the scope to situations where only $P$ visual cues for $N$-speaker audio mixture are available, with $N \geq P$. The extracted speech features are then split into two components: $P$ visually-guided speech features and $N - P$ non-visually-guided speech features. Existing AVSS methods [9, 14, 21, 22, 33] typically aim to separate $N$ individual speakers from the $N$-mixed audio mixture using the corresponding $N$ visual cues. However, in this scenario, these existing AVSS methods encounter the challenge of accurately separating the target speaker without the guidance provided by the corresponding video. In contrast, our separation framework facilitates a comprehensive interaction between visually-guided and non-visually-guided speech features, where the latter serves as the many "missing" parts for frames and speakers as illustrated in Figure 1(b). Specifically, our model effectively enhances the discriminability of separated non-visually-guided speech features by leveraging the complete audio-visual information of other speakers during the separation process. We also consider the scenario where some video frames are partially missing. We randomly select frames for each corresponding video in $N$ speakers with some rates and set those frames to zero, leading to some frame-wise visual information loss. Our model can efficiently compensate for the loss by utilizing both the speech content-related and the speaker-related contextual information. Overall, our model enables a more robust multi-speaker audio-visual separation process, addressing the limitations of the conventional methods.

To sum up, our contributions are as follows:

- We propose a robust multi-speaker audio-visual separation framework that demonstrates superior performance, particularly as the number of speakers in the mixture increases.
- We introduce more challenging scenarios where visual information for specific speakers is absent or visual frames for every speaker are partially missing. Our model achieves the smallest performance gap, further demonstrating its robustness.

## 2 RELATED WORK

### 2.1 Audio-only speech separation

The methods for audio-only speech separation [12, 19, 23–26, 39, 43, 46] can be generally divided into T-domain methods [23–25, 39] and TF-domain methods [43, 46]. Conv-TasNet [24] is an efficient and low-latency end-to-end fully-convolutional time-domain method. DPRNN [23] incorporates dual-path recurrent neural networks to capture long-range information that may be lacking in 1D convolutional networks. Sepformer [22] achieves improved separation performance by leveraging the transformer architecture. However, traditional time-domain methods often face challenges in reverberant conditions due to the absence of explicit frequency-domain modeling. To address this, recent studies have focused on TF-domain approaches [33, 43, 46]. TFPSNet [46] leverages three-path scanning, including T-path, F-path, and TF-path, to extract more comprehensive information. TF-GridNet [43] introduces sub-band and full-band attention LSTM blocks to enhance performance in TF-domain separation tasks.

### 2.2 Audio-visual speech separation

Visual information is naturally aligned with audio in videos, which has become the dominant mode of communication on contemporary social platforms. Many studies [14, 22, 30, 31, 33] have consistently demonstrated that utilizing face-related features, such as still face images and lip movements, can significantly enhance the separation of speech from a mixed audio signal. Convolutional neural networks [7, 14, 27] are widely utilized for effectively handling time-frequency speech features. More recently, the transformer architecture [22, 30, 34] proves to be effective in dealing with time-domain speech features. These approaches all employ the cross-attention mechanism to facilitate the interaction between visual information with time-domain or time-frequency domain speech features. These methods utilize visual features to extract corresponding speech features from the mixture in either a parallel manner [32, 44] or a recursive fashion [45]. Additionally, certain approaches [7, 47] explore two-stage refinement techniques to refine features that may exhibit inconsistencies during the separation process. Our work aims to propose a unified audio-visual speech separation framework that can be more accurate and efficient.

## 3 METHOD

### 3.1 Preliminaries

Expressly, we represent the time-domain speech mixture as $x = \sum_i^N x_i, x_i \in R^{T_x}$, where $N$ denotes the total number of speakers present in the mixture, and $T_x$ represents the time length. Our

basic task is to separate and extract the target audio $x_i$ from the mixed audio $x$ in the time domain, utilizing the provided visual features $V_i$ as a reference for the $i$-th speaker. Considering the situation of visual information deficiency, the number of available visual cues $P$ may be less than the total number of speakers $N$ (i.e., $P \le N$). Following most previous settings [21, 28, 31], our primary focus is on the AVSS task of single-channel audio-visual speech separation, addressing the challenge of separating speech signals from overlapped audio mixtures that contain speeches from different speakers.

The overall pipeline of our model is shown in Figure 2. For clarity, we hypothesize that a 3-speaker mixture with two visual cues was sent into the model. As for the visual input, we send the two visual cues into the visual encoder and obtain visual features with a shape of $R^{P \times T_v \times D}$, where $T_v$ represents the temporal length of visual features, and $D$ represents the feature dimension. As for the speech part, passing through the audio encoder followed by a chunking operation, we obtain speech features represented as $A_m \in R^{N \times S \times L \times D}$. Here, $N$, $S$, $L$, and $D$ denote the number of speakers, the number of chunks, the length of each chunk, and the channel-wise dimension, respectively. In Figure 2, we draw $N = 3$, $S = 3$, $L = 4$ and omit $D$ for better illustration.

## 3.2 Audio and visual encoder

We follow prior research on AVSS methods [22, 31] that utilize a lip embedding extractor. This extractor consists of consecutive frame inputs with length $T_v$ from the lip regions, processed through a 3D convolutional layer followed by an 18-layer ResNet integrated with multi-layer temporal convolutional networks. These modules collectively generate lip motion features $V_i \in \mathbb{R}^{T_v \times D}$ for each speaker $i$, capturing lip movement characteristics in the visual domain. The $P$ speakers' available visual cues form $V \in \mathbb{R}^{P \times T_v \times D}$.

As for the audio part, the audio encoder extracts the features $A_m^o$ from the speech mixture sequence $x \in \mathbb{R}^{T_x}$ using the 1D convolution operation with a kernel size $K$ and stride $K/2$, which is:

$$A_m^o = \text{Conv1D}(x, K, K/2) \in R^{T_a \times D}, \tag{1}$$

where $T_a = \lceil 2 \times T_x / K \rceil$ with proper zero padding, and $D$ is the audio embedding dimension. Due to the typically substantial length of audio features, denoted as $T_a$, computational challenges arise, hindering the learning of short-term, fine-grained features. To address this, a chunk operation is employed to split the audio feature $A_m^o$ into chunks of length $L$ with a hop size of $L/2$. These chunks are then concatenated to form a reshaped 3D audio chunked feature $A_m \in \mathbb{R}^{N \times S \times L \times D}$, where $S$ is the number of generated chunks, and $N$ is the number of speakers in the mixture.

## 3.3 Audio-visual separator

The separator takes the extracted visual features $V \in R^{P \times T_v \times D}$ and audio features $A_m \in R^{N \times S \times L \times D}$ as inputs, and output feature masks $M_i, i \in \{1, \ldots, N\}$ for the $N$ speakers collaboratively with shape $S \times L \times D$. The overall flow for the separator is as follows. The separator in our study is comprised of $B$ blocks, and each block consists of three modules: a local-content attention module, a global content attention module, and a speaker-wise interaction module.

**Fine-grained audio feature extraction**. Speech exhibits a continuous flow lacking explicit segment boundaries, thereby encapsulating intricate and hierarchical content [29]. In contrast, visual cues offer structured data, underscoring the challenge posed by the absence of inherent structure in speech signals. Consequently, the construction of short-time content becomes imperative. We incorporate intra-chunk interactions within each chunk of length $L$ to generate fine-grained information from the short-time features. Feed audio mixture $A_m \in R^{N \times S \times L \times D}$ as input, the process is as follows:

$$\tilde{A}_m = \text{LCA}(A_m[n, s, :, :], \forall n, s), \tag{2}$$

in which LCA represents the local content attention module. By stacking $R$ transformer layers, we enable the interaction among audio tokens of length $L$ within each chunk, capturing both local and contextual dependencies within the speech features.

**Visually guided audio feature enhancement**. After applying the LCA module, we get $N$ speaker features with shape $S \times L \times D$. However, the differentiation among the features for the $N$ speakers is often limited. Despite the facilitation of intra-chunk interactions by the LCA module, the $S$ chunks for each speaker lack comprehensive global information and tend to be similar to each other. In real-world video recordings, there is a strong correlation between a speaker's audio and their lip movements over time. Therefore, the speech and visual features are generally aligned despite different sampling rates. Based on these observations, we introduce the global audio-visual interaction (GAV) module to enhance the disparity for $S$ chunks in each speaker with aligned video features.

Given the temporal length $T_v$ of the visual features and the number of chunks $S$ in the audio features, we ensure that $T_v$ equals $S$ through appropriate padding. Considering that visual cues may be absent for specific speakers, cross-modal interaction is exclusively applied to speakers with available visual cues. The process is illustrated in Part (a) in Figure 2. For the separated speech of shape $S \times L \times D$ accompanied by lip movements of shape $T_v \times D$, we utilize every $1 \times D$ visual frame to interact with each $L \times D$ speech chunk. This can enhance the disparity between $S$ chunks and improve the global feature distribution for $L$ tokens within each chunk. Specifically, given $P$ available visual cues for the $N$ speakers, we split the audio features into two parts of length: $P$ and $N - P$. Moreover, we only use the first part to interact with the corresponding $P$ visual features $V$. This interaction can be mathematically represented as follows:

$$\tilde{A}_{m_1}, \tilde{A}_{m_2} = \tilde{A}_m[: P], \tilde{A}_m[P :],$$
$$\hat{A}_{m_1} = \text{GAV}(V[p, :, l, :], \tilde{A}_{m_1}[p, :, l, :], \forall p, l), \tag{3}$$
$$\hat{A}_m = \text{Concat}(\hat{A}_{m_1}, \tilde{A}_{m_2}),$$

where Concat denotes the concatenate operation. This alignment allows audio features in each chunk to be temporally synchronized with their corresponding visual features, thereby enhancing the diversity of content represented by each audio chunk.

**Inter-chunk global speech interaction**. To facilitate inter-chunk global speech interaction between $S$ chunks, we employ the global content attention module (GCA). The mathematical representation

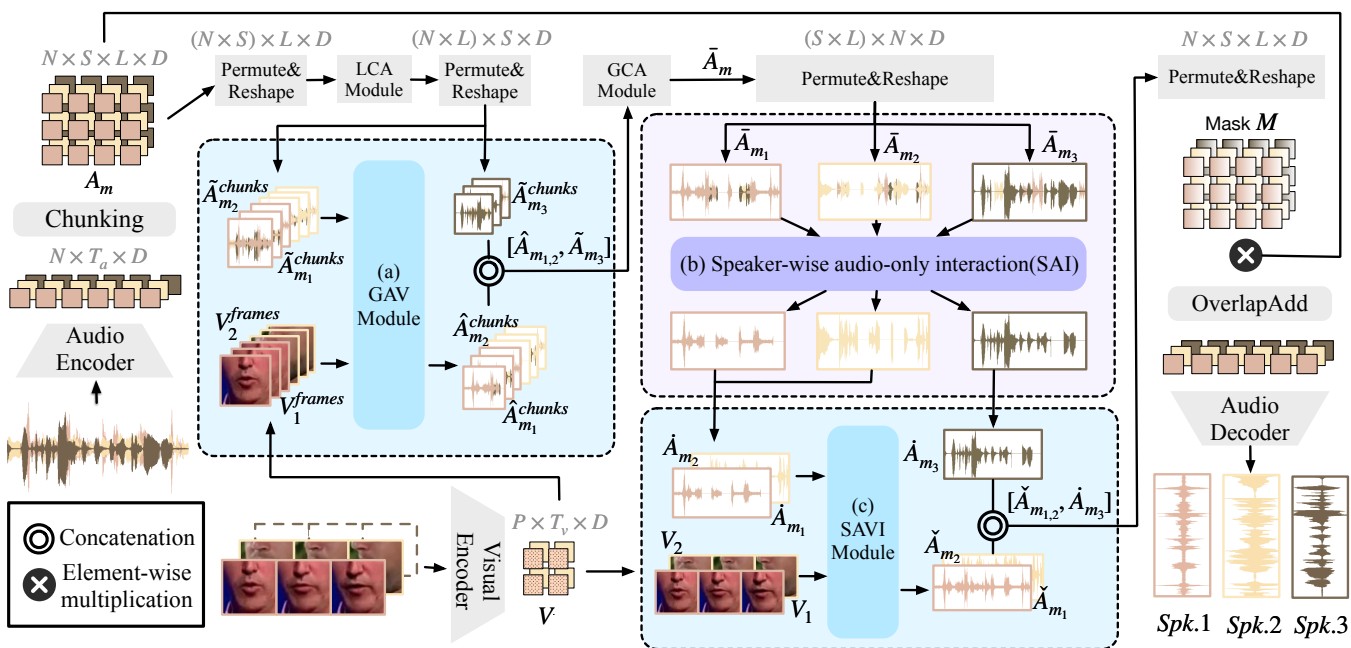

**Figure 2: The overall pipeline of our model. In this figure, we assume that $N = 3$, $P = 2$, $L = 4$ and $S = 3$. The separation block comprises the LCA, GAV, SAI, and SAVI modules. LCA enables effective intra-chunk processing. GCA facilitates inter-chunk processing and enhances inter-chunk disparity by cross-modal visual features. SAI establishes distinctions and correlations between different separated speakers. SAVI further enhances the disparity between hard samples that are similar to each other. Additionally, in scenarios where the available visual information is insufficient for the number of speeches to be separated, we employ a split-and-concat operation to achieve cross-modal interaction.**

of this process is as follows:

$$\bar{A}_m = \text{GCA}(\hat{A}_m[n, :, l, :], \forall n, l), \tag{4}$$

in which $\bar{A}_m$ represents the result of the inter-chunk global speech interaction. By applying the stacked $R$ transformer layers, we enable the inter-chunk speech features to interact globally and capture long-range dependencies within the audio features. This process enhances the understanding and integration of audio information across different chunks.

**Speaker-wise feature interaction.** The previous modules, namely LCA, GAV, and GCA, have been developed to enhance speech content through a blend of fine-grained and coarse-grained interactions, complemented by visual cues. However, these modules have not adequately addressed the precise characterization of speaker attributes for each speaker. As depicted in Part (b) of Figure 2, when handling separated audio features $\bar{A}_m$, there are instances of missing segments or contamination with unrelated segments within each speaker's representation $\bar{A}_{m_1}$, $\bar{A}_{m_2}$, and $\bar{A}_{m_3}$, respectively. Specific audio segments intended for one speaker may erroneously be assigned to others, causing interruptions in the intended speaker's speech. Similarly, segments from other speakers may be incorrectly attributed to a given speaker, leading to contamination. These challenges primarily arise from the limited interaction among speakers. Since the mixture comprises contributions from each speaker, each separated speech component maintains a distinct ratio relative

to the whole and to each other. To address these challenges and enhance the model's robustness in multi-speaker scenarios, we propose a speaker-wise audio interaction (SAI) module followed by a speaker-wise audio-visual interaction (SAVI) module. Starting with the separated speech features $\bar{A}_m$ obtained from Equation (4), the SAI module is employed to facilitate interaction among speaker-specific features more precisely and rigorously. The formulation is as follows:

$$\dot{A}_m = \text{SAI}(\bar{A}_m[:, s, l, :], \forall s, l). \tag{5}$$

The SAI module serves to establish comprehensive global relationships among speakers. It facilitates the reallocation of each component to its corresponding speaker, ensuring accurate assignment of the extracted audio segments. This becomes particularly crucial in scenarios where the distinct separation of speakers within $\bar{A}_m$ lacks distinguishable visual cues. Given that visual information typically improves the accuracy of the speaker-wise attributes, the absence of visual cues can lead to a significant drop in performance. To mitigate this challenge, the SAI module leverages speakers equipped with complete audio-visual information to aid in identifying those solely reliant on audio data. Specifically, we obtain the processed $\dot{A}_m$ after applying the SAI module. Part (c) of Figure 2 demonstrates that the SAI module can filter out unrelated segments for each separated speech from other speakers.

However, when speakers articulate identical sentences with similar pacing and volume, the discrete speech attributes may demonstrate similarity at the semantic level. In such cases, the SAI module

can cause confusion between the two similar separated speeches. To address this issue, it is essential to introduce the speaker-wise audio-visual interaction (SAVI) module to enhance the speaker-related information in each separated speech. The SAVI module plays a crucial role in reducing inherent noises and enhancing the distinctive characteristics specific to each speaker. We firstly divide the separated speaker features $\dot{A}_m$ into two groups: the visually-guided $P$ speakers, denoted as $\dot{A}_{m_1}$, and the non-visually-guided $N - P$ speakers, denoted as $\dot{A}_{m_2}$. The SAVI module is then employed to further enhance the speaker-specific information for the $P$ visually-guided speakers $\dot{A}_{m_1}$. The process for the SAVI can be described as follows:

$$\dot{A}_{m_1}, \dot{A}_{m_2} = \dot{A}_m[: P], \dot{A}_m[P :],$$
$$\check{A}_{m_1} = \text{SAVI}(V[:, s, l, :], \dot{A}_{m_1}[:, s, l, :], \forall s, l), \quad (6)$$

where SAVI represents the global cross-attention process between visual and audio features, which can be described as follows:

$$\text{SAVI}(V, \dot{A}_{m_1}) = \text{Softmax}\left(\frac{V W_Q \cdot (\dot{A}_{m_1} W_K)^{\mathsf{T}}}{\sqrt{d_{\text{head}}}}\right) \dot{A}_{m_1} W_V, \quad (7)$$

where $W_Q$, $W_K$, and $W_V$ represent the weights for query, key, and value embeddings, respectively. Then the processed $P$ visually-guided speaker features in SAVI denoted as $\check{A}_{m_1}$, are concatenated with $N - P$ non-visually-guided speaker features $\dot{A}_{m_2}$ along the speaker dimension, which can be described as:

$$\check{A}_m = \text{Concat}(\check{A}_{m_1}, \dot{A}_{m_2}), \quad (8)$$

where we get the combined $N$ speaker features $\check{A}_m$. Overall, the cohesive operation of the SAI and SAVI modules effectively establishes speaker-specific attributes through inter-speaker interactions and audio-visual cross-modal interactions. These modules ascertain the relative ratio for each speaker to the mixture and to each other, building distinct correlations. They significantly enhance the model's robustness, especially in cases where visual information for certain speakers is completely absent or where visual frames are partially missing.

### 3.4 Audio decoder

To generate the mask $M = \text{OverlapAdd}(\check{A}_m) \in \mathbb{R}^{N \times T_a \times D}$, we apply an overlap-add operation on $\check{A}_m$, which is the inverse process of the chunk operation described in Section 3.2. Next, we perform element-wise multiplication between the output of the audio encoder $A_m$ and the mask $M$, which is given by:

$$A_o = A_m \odot M. \quad (9)$$

Subsequently, we transform $A_o$ back into audio waveforms for each speaker using the transposed convolution version of the encoder. This version employs a kernel size of $K$ samples and a stride size of $K/2$ samples, and the process can be expressed as follows:

$$\hat{x} = \text{TransposedConv1D}(A_o). \quad (10)$$

To address the issue of visual information deficiency, we partition the estimated speech signals $\hat{x}_1$ and $\hat{x}_2$ as $\hat{x}_1, \hat{x}_2 = \hat{x}[: P], \hat{x}[P :]$, along with their corresponding clean audio labels $x_1, x_2 = x[: P], x[P :]$. Due to the potential misalignment between the order of separated features and the order of clean audio labels, we employ distinct training strategies for the video-guided speech signals $\hat{x}_1$

and the non-visually-guided speech signals $\hat{x}_2$. Specifically, for the non-visually-guided speech signals ($\hat{x}_2$) with labels $x_2$, we utilize the Permutation Invariant Training (PIT) strategy [18, 48] combined with the scale-invariant signal-to-distortion ratio (SI-SDR) [37] as the loss function. On the other hand, for the video-guided speech signals ($\hat{x}_1$) with labels $x_1$, we solely employ the SI-SDR loss as the loss function. The total loss function can be described as follows:

$$\mathcal{L} = \mathcal{L}_{SI-SDR}(\hat{x}_1, x_1) + \text{PIT}(\mathcal{L}_{SI-SDR}(\hat{x}_2, x_2)), \quad (11)$$

in which the $\mathcal{L}_{SI-SDR}$ can be described as:

$$\mathcal{L}_{SI-SDR}(\hat{a}, a) = -10 log_{10}\left(\frac{\left\|\frac{\langle \hat{a}, a \rangle a}{\|a\|^2}\right\|^2}{\left\|\hat{a} - \frac{\langle \hat{a}, a \rangle a}{\|a\|^2}\right\|^2}\right). \quad (12)$$

This approach ensures that the model is trained effectively using both visually-guided and non-visually-guided audio features, considering each case's specific requirements.

## 4 EXPERIMENT

### 4.1 Datasets

**VoxCeleb2** [10] comprises over 1 million samples organized by identity labels. It includes 5994 speakers in the training set and an additional 118 speakers in the test set. Each sample contains an utterance with synchronized face tracks. We followed the data generation instructions outlined in [32] to generate the dataset. We randomly selected speech samples from different speakers based on their unique IDs and combined them by adding them together using varying scales. This process resulted in 20,000, 5,000, and 3,000 samples for the training, validation, and test sets.

**LRS3-TED** [2] consists of 433 hours of audio and 151k corresponding video clips extracted from TED and TEDx talks. To assess model generalization, we train them on the VoxCeleb2 dataset and evaluate them on cross-domain LRS3 datasets without fine-tuning. Similar to the VoxCeleb2 mixing process, we generate 660 samples each for 2, 3, 4, and 5 mixture samples as the testing sets.

### 4.2 Implementation details

**Experiment configurations.** The visual frames are sampled at a rate of 25 frames per second (FPS), and the audio data is sampled at a rate of 16kHz. The training length for each mixture is set to 6s. The encoder/decoder kernel size $K$ in Section 3.2 and Section 3.4 is set to 16 with a stride of 8. And the chunk size $L$ and dimension $D$ is set to 160 and 256 respectively. In Section 3.3, the local content attention (LCA) module and global content attention (GCA) module in the separator are stacked with $R$ layers each, which is set to 2. Additionally, all other modules in Section 3.3 are single-layer operations, forming one block for the separator. The one separator block is repeated $B$ times, which is set to 5.

**Optimization.** Following [22], the model is trained using an Adam optimizer [17] with an initial learning rate of $1.5 \times 10^{-4}$. The learning rate will be halved if there is no loss decrease on the validation set for three epochs. The training process stops if there is no loss decrease for five epochs.

**Evaluation.** SI-SDR measures the ratio between the energy of the target signal and that of the errors. We also employ the Perceptual

**Table 1: The testing results for Situation 1. The table shows the separation performance of different AVSS models on the Voxceleb2 and the LRS3 datasets under 2,3,4,5-mix settings, respectively. O.A. represents the overall performance. Red indicates the best performance. Our model performs best among all methods for all 2,3,4 and 5 mixtures.**

| Network | Params | VoxCeleb2 | | | | | | | | | | LRS3 | | | | | | | | | |
|---|---|---|---|---|---|---|---|---|---|---|---|---|---|---|---|---|---|---|---|---|---|
| | | SI-SDR(dB)↑ | | | | | PESQ↑ | | | | | SI-SDR(dB)↑ | | | | | PESQ↑ | | | | |
| | | 2 | 3 | 4 | 5 | O.A. | 2 | 3 | 4 | 5 | O.A. | 2 | 3 | 4 | 5 | O.A. | 2 | 3 | 4 | 5 | O.A. |
| LAVSE [9] | 32.9M | 6.42 | 2.15 | 0.26 | -1.29 | 1.89 | 1.67 | 1.03 | 0.79 | 0.36 | 0.96 | 8.17 | 4.55 | 2.14 | 0.05 | 3.73 | 1.70 | 0.97 | 0.62 | 0.49 | 0.95 |
| AV-ConvTasNet [44] | 16.5M | 10.24 | 5.47 | 2.33 | 1.43 | 4.87 | 2.03 | 1.49 | 1.43 | 1.22 | 1.54 | 12.12 | 6.05 | 3.73 | 2.05 | 5.94 | 2.07 | 1.51 | 1.29 | 1.23 | 1.53 |
| VisualVoice [14] | 77.8M | 10.34 | 5.56 | 2.34 | 1.58 | 4.96 | 2.14 | 1.58 | 1.31 | 1.28 | 1.58 | 13.08 | 6.74 | 4.61 | 1.67 | 7.15 | 2.24 | 1.61 | 1.38 | 1.26 | 1.62 |
| MuSE [32] | 14.3M | 10.70 | 5.85 | 2.89 | 1.97 | 5.35 | 2.10 | 1.57 | 1.38 | 1.24 | 1.57 | 12.42 | 6.51 | 4.21 | 2.45 | 6.40 | 2.15 | 1.58 | 1.36 | 1.25 | 1.59 |
| BFRNet [7] | 53M | 12.47 | 8.35 | 6.23 | 4.64 | 7.92 | 2.38 | 1.89 | 1.60 | 1.57 | 1.86 | 14.35 | 10.54 | 7.72 | 5.23 | 9.46 | 2.39 | 1.76 | 1.62 | 1.56 | 1.83 |
| AV-Sepformer [22] | 26M | 13.23 | 8.89 | 7.31 | 5.48 | 8.72 | 2.45 | 1.88 | 1.66 | 1.61 | 1.90 | 15.42 | 11.12 | 8.73 | 6.68 | 10.49 | 2.54 | 1.94 | 1.76 | 1.69 | 1.98 |
| Ours | 24.3M | 13.94 | 10.06 | 9.21 | 7.60 | 10.20 | 2.55 | 2.02 | 1.86 | 1.69 | 2.03 | 15.77 | 11.73 | 10.60 | 8.65 | 11.69 | 2.61 | 2.07 | 1.92 | 1.75 | 2.08 |

Evaluation of Speech Quality (PESQ) [36] metric to further assess speech quality and intelligibility.

## 4.3 Experiment results

**Training and testing settings.** During the training phase, we generate a dataset with dynamically varying mixture numbers from VoxCeleb2 dataset [10]. This dataset encompasses combinations of 2, 3, 4, and 5 speakers concurrently. Consistent with prior methodologies [7], a distribution ratio of 2:1:1:1 is employed for the respective combinations, thereby fostering equilibrium in performance across diverse compositional scenarios. Additionally, a parameter setting is introduced wherein a 10% probability is designated for the absence of 1-2 visual cues. Subsequent to the training phase, distinct sample sets are generated for each mixture configuration, namely 2, 3, 4, and 5 speakers, during the testing phase. To facilitate a comprehensive evaluation, 3000 samples are specifically generated from the VoxCeleb2 dataset [10], augmented by an additional 660 samples obtained from the LRS3 dataset [2] for each mixture configuration. In order to thoroughly assess the performance of the proposed model and validate its robustness across various scenarios, emphasis is primarily placed on two distinct testing situations.

- Situation 1: Our primary aim is to assess the robustness of our model to variations in the number of speakers, ranging from 2 to 5. Like most previous settings, we send $N$-mix mixtures with corresponding $P$ visual information, in which $P = N$.
- Situation 2: Our primary objective is to assess the robustness of our model in the face of missing visual cues, which encompasses both complete absence of visual cues and partial loss of visual frames. Firstly, to simulate the visual cues absence scenarios, we provide $N$-mix mixtures with corresponding $P$ visual information, in which $P < N$. Secondly, we introduce a random masking process to simulate the scenario of missing visual frames. For each video in the set of $N$ videos, we randomly select a portion of frames based on the specified missing rates and mask them by setting their values to zero.

In Situation 1, we mainly compare the performance of our model with other open-source audio-visual speech separation methods. However, most of these methods present their two-speaker separation results under the two-speaker mixture training setting. To

ensure a fair comparison, we have re-trained these methods using the same training strategy employed in our model, which involves simultaneously training multiple speakers. Note that the difference between a two-speaker training setting and a simultaneously multi-speaker training setting is only the linear increase in the number of speakers in the mixture and the corresponding increase in the number of speakers to be separated during training. To provide further evidence for the superiority of our simultaneous separation framework, we conduct additional comparisons between our methods and selected approaches in the missing visual cues scenarios. The experimental results for the two situations are presented below.

**Experiment results for Situation 1.** The experimental results, presented in Table 1, demonstrate the robust performance of our model across all mixture scenarios. As the number of speakers increases, the task of separation becomes more challenging. On the VoxCeleb2 dataset [10], most methods experience a significant decrease in their SI-SDR performance [37] when the number of speakers to be separated increases from 3 to 4 or from 4 to 5. Our model consistently outperforms other models as the number of mixtures increases, highlighting the effectiveness of our simultaneous separation framework with speaker-wise interactions. The audio-only and audio-visual speaker-wise interaction modules in our model significantly enhance speaker identity in each separated speech and facilitate improved audio-visual fusion. Furthermore, our model's performance is evaluated on the LRS3 dataset [2] without any fine-tuning, demonstrating its generalization and transferability to unseen data. In addition, a consistent positive correlation between the PESQ metric measuring speech quality and the SI-SDR metric can be observed.

**Experiment results for Situation 2.** We test the robustness of our model under conditions where visual information for specific speakers is entirely absent or where visual frames are partially missing, shown in Figure 3 and Figure 4 respectively.

In Figure 3, we compare our results with other methods in scenarios where one corresponding lip movement video is absent. Specifically, we choose AV-Sepformer [22] and BFRNet [7] for comparison. AV-Sepformer is selected because it is also based on the transformer architecture [40], which enables us to demonstrate

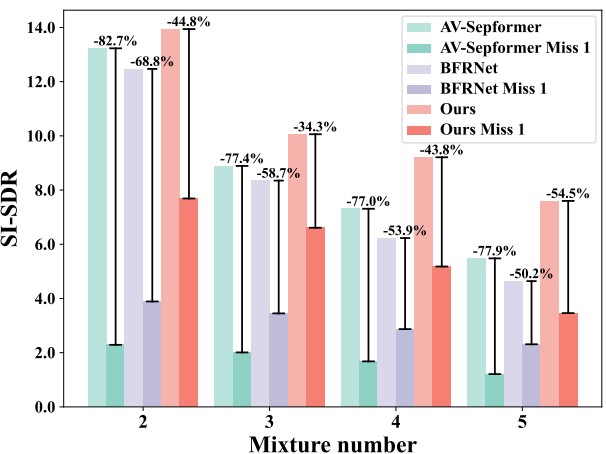

Figure 3: The visual robustness performance with visual absence. We compare the performance of our model with two other methods in 2,3,4 and 5 mixtures, respectively. We evaluate the performance of each model under two conditions: when all visual cues are present and when one visual cue is missing. The picture shows each model's drop rate in performance when one visual cue is missing. We do not include scenarios where more than one visual cue is missing, as they mostly result in negative values.

that the performance improvement is not solely a result of using different backbone architectures. On the other hand, BFRNet is chosen for its specifically designed modules to address the challenges posed by multi-speaker chaos. The results presented in Figure 3 include both the visual-complete scenario in Situation 1 and the part visual-absence scenario for the three methods. Specifically, we demonstrate the visual-absence scenario for situations where only one visual cue is missing in the figure. For example, we demonstrate the performance of 5 separated speech from 5-mix speech mixture with only 4 visual cues. As shown in Figure 3, AV-Sepformer [22] experiences a significant performance drop in the visual-absence scenario, while BFRNet [7] shows a comparatively smoother decline. In contrast, our model consistently outperforms both methods in visual-absence scenarios across all mixtures (2, 3, 4, and 5 speakers). It handles missing visual information effectively, resulting in the smallest performance gap between complete and absent visual scenarios. However, we have yet to present results for scenarios with multiple missing lip movements, which often yield negative values and limit their reference value and interpretability. Bridging the gap between complete and lacking visual information remains an important avenue for future research.

In Figure 4, we present the performance trend of our model as the missing frames rate increases. We compare our model's performance with AV-Sepformer [22] to assess the performance under specific mixture settings. Firstly, we analyze the performance of our model across different mixture settings. It is evident that the performance decreases as the percentage of missing frames increases. Furthermore, as the number of mixtures increases from

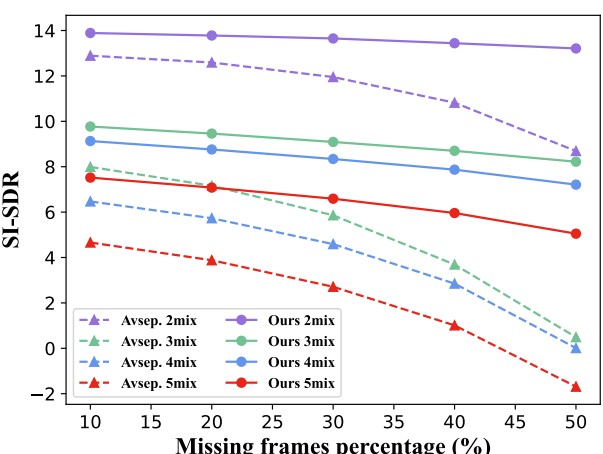

Figure 4: The visual robustness performance with different missing video frame rates. It is important to note that we apply the missing frame rate to all corresponding speakers' video frames. The x-labels of our model represent different missing frame percentages in each corresponding video, and the y-label represents the corresponding SI-SDR metric.

2 to 5, the performance drop becomes more pronounced. Though this can be partly attributed to the overall increase in the number of missing frames, it poses challenges for both speaker extraction and inter-speaker interaction processes. Secondly, we compare our model with AV-Sepformer [22] for each specific mixture setting. Notably, there is a noticeable performance decline as the percentage of missing frames increases. However, our model exhibits a relatively smoother performance drop. This comparison highlights the robustness of our model in scenarios with visual cues part missing. Through this comparison, we observe that our model demonstrates resilience in handling scenarios with missing visual cues, showcasing its ability to maintain performance in challenging conditions.

## 5 ABLATION STUDY

**The effectiveness of each module.** Table 2 demonstrates the effectiveness of each module used in our study. In Row a, we present the basic model [23] consisting solely of LCA and GCA modules, which capture both intra-chunk and inter-chunk correlations of T-domain speech features. By integrating the GAV model within the dual-path framework in Row b, we significantly enhance the speech content disparity for each separated speaker. Compared to the model structure in [22], we make modifications by using fewer layers in each intra-chunk and inter-chunk module, reducing redundancy and increasing the number of cross-modal interactions. As shown in Row b, this modification significantly improves separation performance in mixtures of 3,4,5 speakers by up to 2 dB in SI-SDR [37]. The effectiveness of the single-modal speaker-wise interaction (SI) module is demonstrated in Row c. The SAI module can build the distinctions and correlations among speakers and between each speaker and the whole mixture. It can effectively

**Table 2: Ablation study for different parts in the separation block. We show the effect of GAV, SAI and SAVI modules.**

| No. | GAV | SAI | SAVI | SI-SDR↑ | | | | PESQ↑ | | | |
|-----|-----|-----|------|------|------|------|------|------|------|------|------|
| | | | | 2 | 3 | 4 | 5 | 2 | 3 | 4 | 5 |
| a | ✗ | ✗ | ✗ | 12.76 | 7.96 | 5.72 | 3.89 | 2.29 | 1.72 | 1.44 | 1.31 |
| b | | ✗ | ✗ | 13.42 | 9.21 | 7.65 | 5.82 | 2.47 | 1.90 | 1.69 | 1.57 |
| c | | | ✗ | 13.84 | 9.75 | 8.93 | 7.29 | 2.51 | 1.97 | 1.90 | 1.61 |
| d | | | | 13.94 | 10.06 | 9.21 | 7.60 | 2.55 | 2.02 | 1.86 | 1.69 |

**Table 3: Downstream task performance. Using the open-source pretrained AV-HuBERT model [38], we evaluated the Word Error Rate (WER) metric for the separated speeches. Red indicates the best performance.**

| No. | Method | WER(%)↓ | | | | O.A. |
|-----|--------|------|------|------|------|------|
| | | 2 | 3 | 4 | 5 | |
| a | Clean samples | | | 13.41 | | |
| b | AV-ConvTasNet[44] | 31.14 | 37.17 | 39.31 | 43.35 | 37.74 |
| c | AV-Sepformer[22] | 22.70 | 24.85 | 26.91 | 29.66 | 26.03 |
| d | Ours | 21.25 | 23.49 | 25.65 | 27.68 | 24.52 |

reallocate the mismatching parts in each separated speaker and utilize the visually-guided speech features to help the visual-absence speech features, showing robustness to visual information deficiency. Finally, the speaker-wise audio-visual interaction (SAVI) module in Row d can further enhance the SAI module by providing cross-modal information for speaker-wise distinctions.

**Downstream audio-visual speech recognition performance.** AVSS serves as the front end for various downstream tasks such as speech recognition and speaker diarization. To evaluate the compatibility of separated speech samples with downstream speech recognition tasks, we employ a publicly pretrained model from AV-HuBERT [38] and measure the word error rate (WER) metric on the separated speech. Specifically, we save the separated speech outputs generated by our separation model when provided with 2-mix, 3-mix, 4-mix, and 5-mix inputs, respectively. Due to the absence of text labels in the VoxCeleb2 dataset, we exclusively evaluate performance on the LRS3 dataset. We feed both the separated clean speech and its corresponding video into the audio-visual speech recognition model. As presented in Table 3, the initial clean samples used for mixing exhibit a WER of 13.41. We compare our approach with two classical methods, AV-ConvTasNet [44] and AV-Sepformer [22]. These methods shed light on the developmental progress of speech separation. In Table 3, we notice a direct relationship between the clarity of separated samples (indicated by higher SI-SDR metrics, as shown in Creftab:situation1) and the accuracy of downstream speech recognition tasks, reflected in lower WER. Comparing Row c and Row b, we notice a significant performance improvement resulting from the transition from ConvTasNet to Sepformer. Moreover, our separated speeches achieve the best WER, surpassing others by a large margin. Besides, when comparing Rows b, c, and d

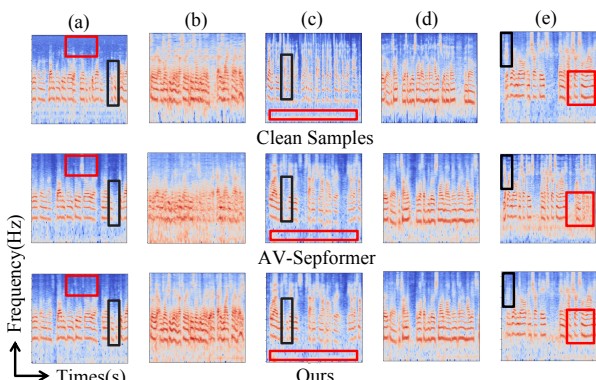

**Figure 5: Visualization results for the separation of a 5-mixture speech audio. Each spectrogram shows time on the horizontal axis and frequency on the vertical axis. The red box indicates time differences within the same row, while the black box highlights frequency differences.**

with Row a, we identify a remaining gap that the speech separation module needs to bridge.

**Qualitative results.** As illustrated in Figure 5, we showcase the separation results for a 5-mixture audio. We visualize the differences between our separated samples, AV-Sepformer, and the original clean samples before the mixing operation. Each spectrogram represents an individual sample, with time on the horizontal axis and frequency on the vertical axis. A spectrogram illustrates the energy distribution across frequencies over time. The red box highlights temporal discrepancies, while the black box indicates frequency distinctions. In Figure 5, we notice disparities in the temporal predictions of the AV-Sepformer model, highlighted in red boxes in Columns (a), (c), and (e). In contrast, our model makes accurate predictions for these segments, closely matching the clean samples depicted in the first row. Moreover, our model adeptly predicts missing or noisy segments across both high and low frequencies, as illustrated by the black boxes in Columns (a), (c), and (e). Additionally, in Column (b), we observe chaotic energy distribution in specific segments of the AV-Sepformer predictions, indicating a more pronounced decline in performance.

## 6 CONCLUSION

In this paper, we propose a robust simultaneous multi-speaker audio-visual separation framework that is robust to both multi-speaker scenarios and missing visual cues. Our framework utilizes cross-modal audio-visual interaction modules to enhance the distinctions between speakers in terms of both speech content and speaker attributes. Additionally, an inter-speaker interaction module is employed to establish relationships among speakers and between individual speakers and the entire audio mixture. It effectively establishes the connection between visually-guided separated speakers and non-visually-guided separated speakers. The framework achieves state-of-the-art results in settings involving 2, 3, 4, and 5 speaker mixtures and significantly reduces the performance gap in scenarios where visual cues are absent.

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
