# OpenReview forum: "RAVSS: Robust Audio-Visual Speech Separation in Multi-Speaker Scenarios with Missing Visual Cues"
_acmmm.org/ACMMM/2024/Conference — MM2024 Poster_

### Official Review · Reviewer_hfVz · 2024-05-20

**Rating:** 5
**Confidence:** 4

**Summary:**

This paper addresses the challenge of robustness in scenarios involving multiple speakers and missing visual cues for audio-visual separation. To tackle this, the authors introduced audio-visual interaction modules that leverage the correlation between speech and lip movements. Specifically, they utilize content and speaker-wise interaction modules to address cases where visual cues of speakers are missing using complete audio-visual information.

**Strengths:**

This paper is well-motivated and exhibits clear and focused writing.
It's good to see that the authors effectively introduce each module, providing insights into their design rationale and specific functions. The model design is aligned with the stated motivation, ensuring coherence throughout.
Furthermore, the experimental design revolves around assessing the system's robustness to variations in the number of speakers and missing visual cues, showcasing a thoughtful approach to evaluation.

**Limitations:**

1. As the authors claimed in the paper, the feature after SAI and SAVI module will contain the distinctive characteristics specific to each speaker. A direct approach to validate this claim would be through speaker verification tests. I recommend that the authors conduct experiments to demonstrate the capability of these features in speaker verification tasks.

2. Nowadays audio-only separation can also perform well. Can you compare with audio-only separation systems when all visual cues in your model are missing?

3. Considering recent advancements, it would be beneficial to include comparisons with newer models such as the Audio-visual TF-GridNet [1].

[1] Pan, Z., Wichern, G., Masuyama, Y., Germain, F. G., Khurana, S., Hori, C., & Le Roux, J. (2023, December). Scenario-Aware Audio-Visual TF-GridNet for Target Speech Extraction. In 2023 IEEE Automatic Speech Recognition and Understanding Workshop (ASRU) (pp. 1-8). IEEE.

**Suitability:**

3

---

### Official Review · Reviewer_X2Bz · 2024-05-20

**Rating:** 4
**Confidence:** 3

**Summary:**

This paper addresses the problem of separating speech from multiple speakers using both audio and visual modalities. While existing approaches using audio-visual modalities handle this problem for up to two speakers, the authors propose an end-to-end framework that solves the problem for up to five speakers. Their framework also demonstrates robustness when visual information is incomplete or missing.

**Strengths:**

1. **Clarity and Detail**: The paper is well-written, providing clear and detailed explanations of the proposed methods and the problem being addressed.
2. **Problem Statement and Motivation**: The authors clearly define the problem and provide strong motivation for their work, thoroughly reviewing relevant literature to position their contributions within the broader research context.
3. **Innovative Contributions**: The paper introduces novel attention mechanisms to capture local, global, and audio-visual interaction features, significantly advancing the state-of-the-art in multi-speaker speech separation.
4. **Evaluation**: The proposed method is evaluated on challenging datasets ([VoxCeleb2](https://www.robots.ox.ac.uk/~vgg/data/voxceleb/vox2.html) and [LRS3](http://www.robots.ox.ac.uk/~vgg/data/lip_reading/lrs3.html)), demonstrating its effectiveness in separating mixtures with up to five speakers and showing robustness to missing visual cues.

**Limitations:**

1. **Inconsistency in Notation**: The variable \( N \) denotes the number of speakers and \( P \) denotes the number of available visual cues. However, the notation does not consistently guarantee clarity throughout the paper.
2. **Reliance on arXiv References**: Many references are from arXiv, which are not peer-reviewed. This raises concerns about the reliability and validation of the referenced works.
3. **Small Margin of Improvement**: In Table 1, the performance differences between the proposed approach and existing methods (e.g., AV-Sepformer) are relatively small for some metrics (e.g., SI-SDR and PESQ) on the VoxCeleb and LRS3 datasets. Without running the approaches multiple times and performing statistical analysis, it is challenging to conclusively determine superiority.
4. **Lack of Detailed Explanation for LCA and GCA**: The paper lacks detailed descriptions of the Local-Content Attention (LCA) and Global Content Attention (GCA) modules. The authors simply state they use "stacking RR transformer layers" without further elaboration. Additionally, the ablation study does not evaluate the individual contributions of LCA and GCA, leaving their impact unclear.
5. **High Number of Parameters**: The proposed approach has 24.3 million parameters, which is high for real-time and practical applications. Given that speech separation is often a front-end task for other speech processing applications, the model's complexity might limit its usability in practical scenarios.

**Suitability:**

3

---

### Official Review · Reviewer_AGRh · 2024-05-23

**Rating:** 3
**Confidence:** 3

**Summary:**

This paper introduces RAVSS, a robust framework for audio-visual speech separation that excels in multi-speaker scenarios, even with missing visual cues. Unlike existing methods that sequentially isolate speakers and struggle with performance drops as the number of speakers increases, RAVSS performs simultaneous separation and utilizes speaker-wise interactions to distinguish and correlate among speakers effectively. Experimental results on the VoxCeleb2 and LRS3 datasets demonstrate that RAVSS achieves state-of-the-art performance in separating mixtures of up to five speakers, maintaining robustness in scenarios with missing or partial visual information.

**Strengths:**

The paper includes extensive evaluation results. it also includes qualitative samples to show how the separation works well.

**Limitations:**

The writing of this paper needs to improve. It is not clear to show the novelty and strengths of this paper. For example, the contribution "We propose a robust multi-speaker audio-visual separation framework that demonstrates superior performance, particularly as the number of speakers in the mixture increases" does not include concrete instances. Figure 2 is complex but it is not clear the key contribution and module in the model. There are several fusion modules but how they are unique and effective is not well presented.
Figure 5 c and d do not have annotations. Furthermore, it makes more sense to show how a segment speech of one person is incorrectly mapped to another speech.
The paper does not show the novelty to solve the problem. It introduces several modality fusion modules but does not show how they are different from previous work. There are no figures on the details inside several proposed modules.

**Suitability:**

2

---

### Official Review · Reviewer_8Nn8 · 2024-05-24

**Rating:** 5
**Confidence:** 4

**Summary:**

The paper focuses on target speaker extraction using visual cues, even in scenarios where some visual cues are missing or incomplete. The authors propose a novel RAVSS network capable of handling scenarios with 2-5 mixed speakers, achieving state-of-the-art results. Additionally, they demonstrate the robustness of their network in the absence of visual cues.

**Strengths:**

1.This paper is novel enough to address missing visual cues problems in target speaker extraction tasks using visual cues. In the experimental section, it compares various existing methods in Table 1 with exhaustive experiments in scenarios with two to five speakers, and demonstrates that its method works best in normal audio-visual scenario.
2.The most important comparison is in Figure 3 where visual cues are missing, and the results in the figure demonstrate the strong robustness of this paper to the case of missing visual cues, which is a very significant advantage over previous work.
3.The interpretability of the results is high. Figure 5 clearly demonstrates the advantages of the proposed method compared to others. Across low, medium, and high frequencies, RAVSS more effectively removes extraneous noise.

**Limitations:**

1.The paper is not clear enough in its description of specific modules, such as LCA, GAV and GCA etc., and does not give detailed structures or descriptions, which only seem to be presumed to be some kind of attentions or transformers in the paper's description. While the supplementary material may be given in the code, it still needs to be described clearly in the paper.
2.The RAVSS network seems to require prior knowledge of which speaker's visual cues will be missing at the input, rather than learning this automatically. For instance, in the 3mix scenario, if the presence or absence of visual cues is unknown during the inference stage, will RVASS assume P=N by default? In this case, does the specially designed approach of RAVSS for handling situations where P<N still have practical significance?

**Suitability:**

3

---

### Meta-Review · Area_Chair_eJg4 · 2024-07-03

**Recommendation:** Accept (Poster)
**Confidence:** 4

**Metareview:**

The paper proposes a new simultaneous multi-speaker separation framework that considers speaker-wise interactions. The paper received 2 WAs, 1 BA, and 1 BR. The BR reviewer is mainly concerned that the paper's presentation is not clear. Otherwise, the reviewers liked that the problem and method have good motivation, the method is robust, and the evaluation is thoughtful. The AC recommends acceptance and thinks the presentation is addressable and urges the authors to improve paper writing, especially regarding the modules LCA and GCA.